

# Tetrandrine alleviates cerebral ischemia/reperfusion injury by suppressing NLRP3 inflammasome activation via Sirt-1

Jun Wang[1,2,*], Ming Guo[1,*], Ruojia Ma[1], Maolin Wu[1] and Yamei Zhang[1]

[1] Department of Cardiology, Zhejiang Xiaoshan Hospital, Hangzhou, Zhejiang Province, China
[2] Department of Acupuncture, Zhejiang Provincial Integrated Chinese and Western Medicine Hospital, Hangzhou, Zhejiang Province, China
[*] These authors contributed equally to this work.

Corresponding author
Yamei Zhang, zymei77@163.com

## ABSTRACT

**Background & Aims**. Tetrandrine (Tet) has been reported to have anti-inflammatory effects and protect from the ischemic strokes. The NLRP3 inflammasome plays a key role in cerebral ischemia/reperfusion (I/R)-induced inflammatory lesions. However, the molecular mechanisms of Tet related to the progression of cerebral ischemia are still unclear. Therefore, the aim of this study was to investigate the possible effects of Tet on cerebral ischemia and the related mechanisms involved in NLRP3 inflammasome.
**Methods**. C57BL/6J mice used as a cerebral I/R injury model underwent middle cerebral artery occlusion (MCAO) for 2 h following reperfusion for 24 h. Tet (30 mg/kg/day, *i.p.*) was administered for seven days and 30 min before and after MCAO. Their brain tissues were evaluated for NLRP3 inflammasome and Sirtuin-1 (Sirt-1) expression. An intracerebroventricular injection of Sirt-1 siRNA was administered to assess the activation of the NLRP3 inflammasome.
**Results**. Tet significantly reduced the neurological deficits, infarction volume, and cerebral water content in MCAO mice. Moreover, it inhibited I/R-induced over expression of NLRP3, cleaved caspase-1, interleukin (IL)-1β, IL-18, and Sirt-1. Sirt-1 knockdown with siRNA greatly blocked the Tet-induced reduction of neurological severity score and infarct volume, and reversed the inhibition of NLRP3 inflammasome activation.
**Conclusion**. Our results demonstrate that Tet has benefits for cerebral I/R injury, which are partially related to the suppression of NLRP3 inflammasome activation via upregulating Sirt-1.

## INTRODUCTION

Ischemic strokes are a leading cause of disability and death worldwide, and place a heavy burden on patients and society (*Hou et al., 2019*). Although promising researches on the mechanisms of stroke have been conducted in recent years, neuroprotective strategies for clinically application are still lacking. Therefore, seeking effective candidates

has become the focus of research. The pathophysiological mechanisms of stroke, especially cerebral ischemia/reperfusion (I/R) injury, are involved in energy metabolism impairment, glutamate/neurotoxin release, autophagy, and inflammation (*Guyot et al., 2000*). Inappropriate immune response and the inflammatory cascade have been increasingly recognized as important pathological factors influencing I/R injury. The early phase of I/R activates cerebral immune cells, such as microglia, which are responsible for the generation of inflammatory mediators and immunity activation (*Schmidt et al., 2016*).

It is well established that inflammasomes are involved in several brain disorders, such as Alzheimer's disease, Parkinson's disease, epilepsy, and stroke (*Aminzadeh et al., 2018*; *Slowik et al., 2018*). The NOD-like receptor pyrin domain-containing 3 (NLRP3), is known as a major component of the inflammasome. It is an intracellular multiprotein signaling complex that includes the NLRP3 scaffold, the adaptor protein PYCARD/ASC, and caspase-1. Once activated, the inflammasome prompts the activation of caspase-1 and converts pro-interleukin (IL)-1β and pro-IL-18 into their mature forms, which can aggravate inflammatory reactions (*Mehto et al., 2019*). Various stimuli, such as the release of triphosphate (ATP) by nigericin and injury cells, can trigger the activation of the NLRP3 inflammasome (*Chen & Chen, 2018*). NLRP3 and the inflammasome pathways have been shown to be related to inflammation-associated diseases such as atherosclerosis, type II diabetes mellitus and cancer (*Karasawa & Takahashi, 2017*; *Rovira-Llopis et al., 2018*; *Wei et al., 2014*). Accumulating evidence indicates that the NLRP3 inflammasome plays a decisive role in the development of cerebral I/R injury (*Qiu et al., 2016*; *Wang et al., 2015*). In addition, microglia, innate immune cells in the brain, express NLRP3 to mediate inflammatory cytokine production. The NLRP3 pathway is therefore considered a beneficial target for cerebral I/R injury.

Tetrandrine (Tet) is a unique alkaloid extracted from the root of a Chinese herb called Radix *Stephania tetrandra*. Tet has drawn considerable attention for its anti-tumor (*Chen, Chen & Tseng, 2009a*; *Chen, Chen & Tseng, 2009b*; *Chen et al., 2009*), anti-inflammatory (*Li et al., 2018*), and analgesic activity (*Zhang & Fang, 2001*), and has been widely used for these purposes since ancient times. Studies have shown Tet to be a neuroprotective agent against ischemic stroke (*Sun & Liu, 1995*; *Ruan et al., 2013*). In a middle cerebral artery occlusion (MCAO) mouse model, treatment with tetrandrine was found to reduce infarct volume and brain water content (*Ruan et al., 2013*). Furthermore, Tet suppresses the production of pro-inflammatory mediators in ischemia in vivo (*Chen, Chen & Tseng, 2009a*; *Chen, Chen & Tseng, 2009b*; *Chen et al., 2009*). However, the possible impact of Tet on the NLRP3 pathway in mice with cerebral ischemia has not yet been reported.

Sirtuin-1 (Sirt-1), a member of the sirtuin enzyme family, which includes seven proteins, is distributed in the central nervous system (CNS) of mammals. Several studies have indicated a significant role of Sirt-1 in the neuroprotective mechanism (*Kaur et al., 2015*; *Wang et al., 2019a*; *Wang et al., 2019b*). At the molecular level, it promotes interaction with DNA and several substrates deacetylated and downregulates NLRP3 inflammasome activation in renal epithelial cells (*Chou et al., 2019*). The role of Sirt-1 expression in regulating the activition of the NLRP3 inflammasome induced by cerebral ischemia remains to be established.

In this study, we investigated whether Tet administration has the neuroprotective effects on cerebral I/R injury in an MCAO mouse model. In addition, we sought to reveal the relationship between this neuroprotective effect and Sirt-1-mediated NLRP3 inflammasome activation *in vivo*.

## MATERIALS & METHODS

### Animals

Male C57BL/6J mice (weight $25 \pm 3$ g) were purchased from Zhejiang Academy of Medical Sciences, Hangzhou, China with experimental animal use license SYXK 2014-0008. All mice were maintained in a 12 h light-dark cycle, $22-25$ °C and relative humidity $55 \pm 5\%$ environment, and were free access to water. All experimental animals were performed in strict compliance with the National Institutes of Health Guide for the Care and Use of Laboratory Animals. Procedures were approved by the Institutional Animal Care and Use Committee of the Zhejiang Academy of Medical Sciences. The experimental procedures were approved by the Ethics Committee of Laboratory Animal Care and Welfare, Zhejiang Academy Medical Sciences, with the proved number 2018-143.

### Experimental protocols

A total of 45 mice were randomly divided into three groups ( $n = 6$ for TTC staining; $n = 6$ for behavioral test and protein extraction; $n = 3$ for immunofluorescence analysis): sham operation (sham, $n = 15$), MCAO with vehicle (vehicle, $n = 15$) or tetrandrine (MCAO + Tet, $n = 15$). Tetrandrine (Sigma-Aldrich, St. Louis, MO, USA, PHL89321) was freshly prepared in normal saline. The mice in the MCAO + Tet group received a Tet dose of 30 mg/kg intraperitoneally (*Ruan et al., 2013*) once a day for seven days before surgery and 30 min before and after inducing ischemia. The vehicle group was injected with an equal volume of normal saline. The mice were assigned neurological severity scores 24 h after MCAO surgery. All mice were then anesthetized with ketamine (100 mg/kg, *i.p.*) and sacrificed by cervical dislocation. The brains were harvested to measure cerebral infarct volume and brain water content (Fig. 1A). In addition, an intracerebroventricular injection of Sirt-1 small interfering RNA (siRNA) was administered to C57BL/6J mice to inhibit cerebral Sirt-1 expression. Another 48 mice were randomly divided into two groups: 18 mice for scrambled siRNA injection and 30 mice for Sirt-1 siRNA injection. After 48 h, all 48 mice were subjected to MCAO. Furthermore, a Tet group ($n = 15$) and a Tet + Sirt-1 siRNA group ($n = 15$) were treated with Tet as described above, and mice in a Sirt-1 siRNA group ($n = 15$) were received an equal volume of normal saline intraperitoneally. Three mice with scrambled siRNA injection and MCAO were used as controls for Western blot analysis. At the end of the experiment, the surviving animals were sacrificed.

### Establishing the cerebral I/R injury induced by MCAO

The operating procedure for transient focal cerebral ischemia was previously described by *Liu et al. (2018)*. Briefly, the mice were deeply anesthetized, and a midline incision in the neck was made to expose the right external carotid artery (ECA) and the right internal carotid artery (ICA). Silicone-coated nylon monofilament (0.28 mm in diameter) was

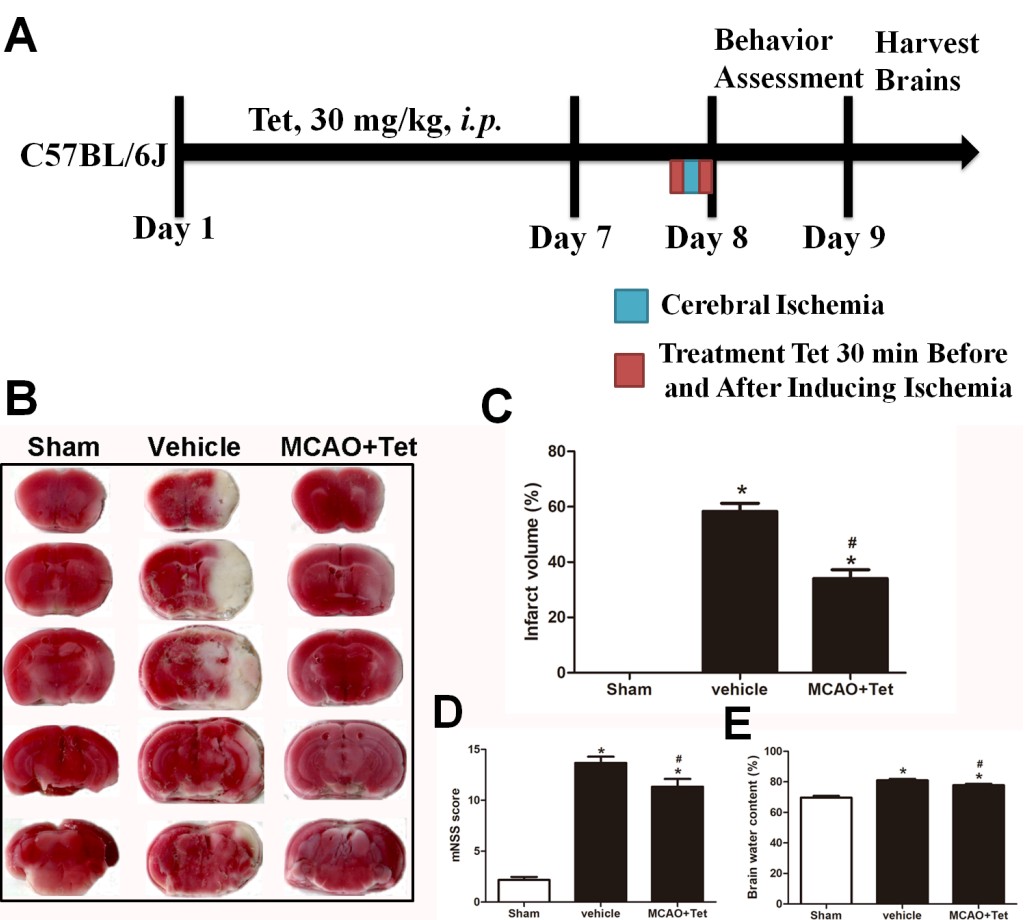

**Figure 1** **Tetrandrine (Tet) alleviated middle cerebral artery occlusion (MCAO)-induced injury in mice.** (A) Illustration of experimental schedule. The C57BL/6J mice received Tet (30 mg/kg, *i.p.*) or vehicle for 7 days. Then, the mice were subjected to MCAO, followed by Tet treatment 30 min before and after surgery. The neurobehavioral outcomes and infarct volumes were assessed on day 9. The representative images of TTC-stained brain Section B and quantification the infarct area (C) were shown. Neurological scores (D) and brain water content (E) were measured after cerebral ischemia. Values are mean ± SEM, ($n = 6$ per group). $^*P < 0.05$ *vs.* Sham; $^\#P < 0.05$ *vs.* vehicle.

gently inserted from the ECA into the ICA lumen until a 15–19 mm intraluminal thread obstructed the origin of the MCA for 90 min. Then, the nylon was withdrawn to restore blood flow. In the sham group, the mice underwent the same operating procedure without thread insertion.

## Assignment of neurologic severity score

Twenty-four hours after the ischemic operation, to assess neurological defects, modified neurological severity scores (mNSS) were assignedby a blinded investigator (*Chen et al., 2017*). The tests evaluated the motion, sensation, reflex, muscle state, abnormal movement, vision, tactile sense and balance systems of the mice. The scores were assigned on a scale from 0 to 18, where 0represented no evident neurological deficits, and 18 represented severe deficits.

## Measurement of brain water content

After the neurological functions were evaluated, the mice were euthanized, and their brains were removed immediately. The cerebral cortex (at two mm around the craniotomy) was isolated, and blood and cerebrospinal fluid were removed with filter paper. After the wet weight was measured using an analytical balance, the samples were dried in an oven at 100 °C for 24 h. The dry weight was then measured with an analytical balance. The brain water content was calculated according to as the formula % = (wet weight - dry weight)/wet weight ×100.

## Measurement of cerebral infarct volume

The cerebral infarct volume was determined using 2,3,5-triphenyltetrazolium chloride (TTC) staining as previously described (*Liu et al., 2018*). The brains were cut into five coronal sections, which were incubated in 2% TTC at 37 °C for 30 min. All slides were then fixed in 4% paraformaldehyde buffer for 24 h. The infarct and total hemispheric areas were measured using the ImageJ analysis software (National Institutes of Health, Bethesda, MD, USA). The ischemic volume was calculated as the percentage of cerebral ischemic volume to the total volume of the sections.

## Immunofluorescence analysis

Ischemic hippocampal tissue was fixed in formaldehyde before being embedded in paraffin and cut into 4 $\mu$m sections. The samples then underwent deparaffinization with dimethyl benzene, gradient alcohol dehydration, and antigen retrieval according to the citric acid buffer/microwave protocol. The sections were incubated overnight with primary antibodies for Iba-1 (diluted 1:200; Abcam, Cambridge, UK) and NLRP3 (diluted 1:200; Invitrogen, Grand Island, NY, USA) at 4 °C. The slides were then washed with phosphate-buffered saline (PBS) and incubated with goat anti-rabbit IgG antibody (diluted 1:200; Invitrogen, Grand Island, NY, USA) at room temperature for 1 h. Following staining with 4′,6-diamidino-2-phenylindole (DAPI; diluted 1:300; Molecular Probes/Invitrogen Life Technologies, Eugene, OR, USA), the slides were examined with a fluorescence microscope (Leica Microsystems, Wetzlar, Germany). Image analysis was performed using Image J (National Institutes of Health, Bethesda, MD, USA). The density of Iba-1-positive cells and celsl with Iba-1 localization of NLRP3 was measured (cells/mm$^2$). One section per mice and three mice per group were evaluated.

## Determination of inflammatory cytokines

Inflammatory factors in cerebral tissue were determined with enzyme-linked immunosorbent assay (ELISA) kits (Anogen, Mississauga, Ontario, Canada) according to the manufacturer's instructions. Briefly, IL-1β and IL-18 in both standards and samples were performed with monoclonal anti-mouse IL-1β and IL-18 as primary antibodies. All OD values were converted into corresponding concentration values.

## Intracerebroventricular injection of Sirt-1 siRNA

Sirt-1 siRNA (sense primer, 5′-GCAGAUUAGUAAGCGUCUUTT-3′; antisense primer, 5′-AAGACGCUUACUAAUCUGCTT-3′) was designed and synthesized by
GenePharma Corporation (Shanghai, China). A scrambled siRNA (sense primer, 5′-GCGCCAGUGGUACUUAAUAUU-3′; antisense primer, 5′-UAUUAAGUACCACUGGC GCUU-3′) was synthesized without a target sequence for the control. An intracerebroventricular injection of siRNA was administered according to the method described by *Wang et al. (2019a)* and *Wang et al. (2019b)*. The mice were deeply anesthetized. Hair was removed from the center of the head after swabbing with povidone iodine and 75% ethanol. The mice were then placed on a stereotaxic apparatus. A 25 μL Hamilton syringe was fixed on the stereotaxic apparatus and inserted perpendicularly at 1.0 mm posterior to the bregma and 2.0 mm lateral to the midline and to a depth of 3.5 mm beneath the surface of the skull. Next, 10 μL of Sirt-1 siRNA diluent (2 μg/ μL) was injected into the right lateral ventricle at a rate of 1μL/min. Upon completion, the needle was gently withdrawn. Forty-eight hours later, the mice were subjected to MCAO.

## Western blot analysis

Protein was extracted from brain tissue with radioimmunoprecipitation assay lysis buffer (Dallas, TX, USA), and a total of 30 μg of protein was separated by sodium dodecyl sulfate-polyacrylamide gel electrophoresis. The protein was then transferred to a nitrocellulose membrane and blocked in 10% skimmed milk at room temperature for 1 h. The membranes were incubated overnight with primary antibodies against NLRP3 (1:500; Invitrogen, Grand Island, NY, USA), cleaved caspase-1 (1:500, Abcam, Cambridge, UK), Sirt-1 (1:300, Cell Signaling Technology, Danvers, MA, USA), and glyceraldehyde 3-phosphate dehydrogenase (GAPDH; 1:5,000; Bioworld Technology, St Louis Park, MN, USA) at 4 °C. The membranes were then washed with PBS and incubated with horseradish peroxidase-conjugated IgG (1:5,000, Cell Signaling Technology, Danvers, MA, USA) secondary antibodies at room temperature for 1 h. The protein bands were visualized using an ECL Western Blotting Detection System (Millipore, Billerica, MA, USA) and normalized for GAPDH expression.

## Statistical analysis

The results were expressed as means ± standard error of the mean (SEM). Statistical analysis was performed by one-way analysis of variance (ANOVA) followed by Newman-Keuls multiple comparison tests. A value of $P$ less then 0.05 was considered statistically significant.

# RESULTS

## Tet ameliorated cerebral I/R injury

To explore the neuroprotective effects of Tet against cerebral I/R injury, cerebral infarct volume, mNSS scores, and brain water content were examined. As shown in Fig. 1B and 1C, we observed that infarct volume in the MCAO + Tet group was clearly smaller than in the vehicle group. Similarly, the Tet treatment significantly improved the neurological severity scores and decreased the brain water content in the MCAO + Tet group compared with the vehicle group (Fig. 1D and 1E). Taken together, these data indicated that the Tet treatment had protective effects on cerebral I/R injury.
## Tet inhibited the activation of the NLRP3 inflammasome in the hippocampal microglia of ischemic brains

To investigate the influence of Tet on NLRP3 inflammasome in hippocampal microglia, immunofluorescence staining was used. The results showed that the density of Iba-1-positive cells in the hippocampus region, corresponding to the number of microglia, was higher in the vehicle group than in the sham group (Figs. 2B, 2F, and 2M). Stronger and more extensive staining for NLRP3 appeared in the hippocampus region of mice in the vehicle group compared with those in the sham group (Figs. 2A and 2E), and their microglia were activated, showing a dramatic increase in positive staining for NLRP3 (Figs. 2D, 2H, and 2N). However, the Tet treatment greatly reduced the number of Iba-1-positive cells and the percentage of cells with Iba-1 localization with NLRP3 in the MCAO + Tet group compared with the vehicle group (Figs. 2E–2N).

## Tet inhibited NLRP3 -derived inflammation and upregulated Sirt-1 expression in cerebral I/R injury

To clarify the protective effects of Tet related to NLRP3 -derived inflammation, we used Western blot to detect the protein levels of NLRP3 and cleaved caspase-1, and ELISA to detect IL-1β and IL-18 levels in brain tissue. The data showed that NLRP3, cleaved caspase-1, IL-1β, and IL-18 levels were significantly elevated in the vehicle group compared with the sham group. Importantly, these increases were dramatically inhibited in the MCAO + Tet group compared to the vehicle group (Figs. 3A–3E). To determine the effect of Tet on Sirt-1 expression in response to cerebral I/R injury, we evaluated the protein level of Sirt-1. The protein levels of Sirt-1 were significantly lower in the cerebral I/R-induced group than in the sham group. Treatment with Tet upregulated the expression of Sirt-1 in the MCAO + Tet group compared to the vehicle group (Fig. 3F and 3G). These results suggested that I/R-induced cerebral injury improvement through Tet was related to the inhibition of NLRP3-regulated release of inflammatory cytokines and the upregulation of Sirt-1.

## Tet suppressed NLRP3 inflammasome activation through Sirt-1

To elucidate the molecular mechanisms of Sirt-1 on the activation of NLRP3 inflammasome components in response to cerebral I/R injury, cerebral Sirt-1 was partially knocked down by siRNA. Compared with the Tet group, Sirt-1 siRNA pretreatment increased cerebral infarct volume and neurological severity score. There were no significant differences in infarct volume and neurological score between the Sirt-1 siRNA group and the Tet + Sirt-1 siRNA group (Figs. 4A–4C). In addition, Sirt-1 siRNA significantly reduced the protein level of Sirt-1 in the mice that underwent MCAO (Figs. 4D–4E). These results indicated that Sirt-1 played a role in Tet-mediated neuroprotective effects under I/R stimulation. Furthermore, we found that Sirt-1 siRNA markedly increased the number of Iba-1-positive cells and the percentage of NLRP3 in hippocampal microglia in the Sirt-1 siRNA group compared to the Tet group (Figs. 5A–5E). Similarly, Sirt-1 siRNA effectively upregulated IL-1β and IL-18 in brain tissue compared to the Tet group (Fig. 5F and 5G). Moreover, Tet-induced suppression of NLRP3-derived inflammation, including NLRP3, IL-1 β, and IL-18, was significantly abolished by partial Sirt-1 knockdown (Figs. 5A–5G). Overall, these

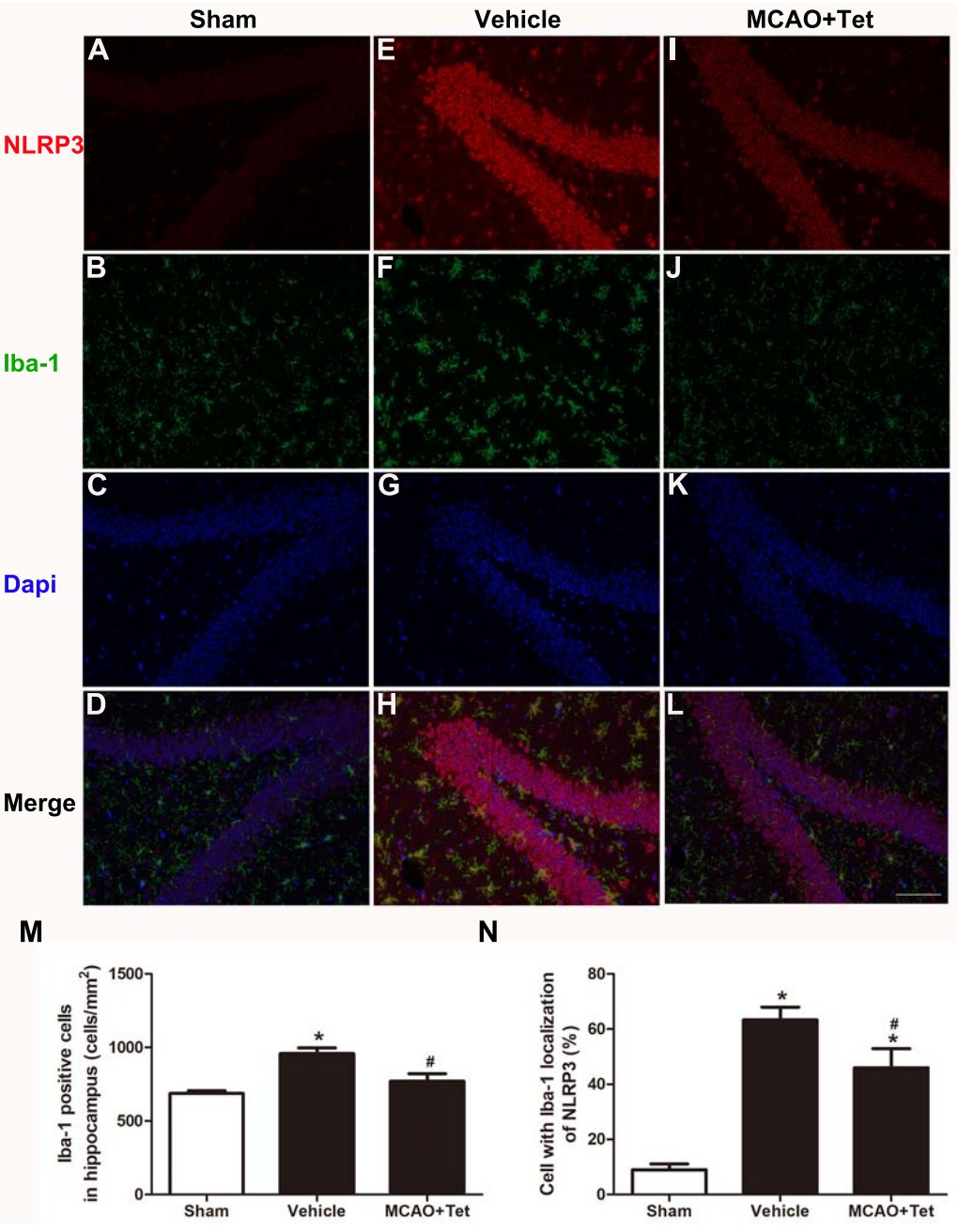

**Figure 2** **Tetrandrine inhibited the activation of the NLRP3 inflammasome in hippocampal microglia of ischemic brain.** Brain sections were stained with DAPI (blue), as well as Iba-1 (green) or NLRP3 (red) to monitor NLRP3 accumulation. (A–L) Representative immunofluorescent staining in the hippocampus of mice. Scale bar, 100 $\mu$m. Iba-1 positive cells in the hippocampus (M) and percentage of cells with Iba-1 localization of NLRP3 (N) were quantified. Tet, Tetrandrine.

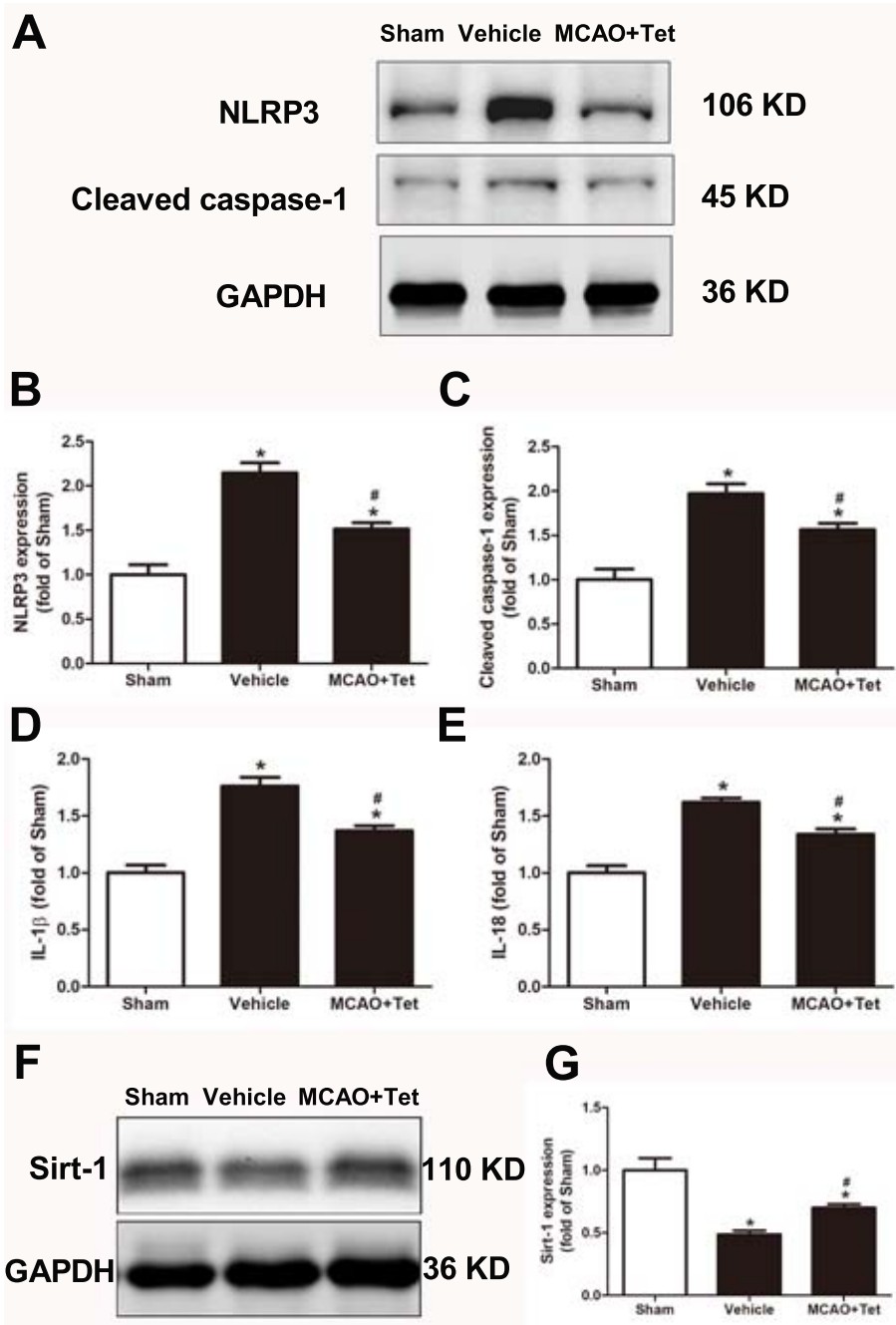

**Figure 3** **Tetrandrine inhibited NLRP3 -derived inflammation and upregulated Sirt-1 expression in ischemic brain in mice.** (A) Protein levels of NLRP3 and cleaved caspase-1 in brain tissues were measured by western blots and were normalized to GAPDH. The bar graphs show that Tet treatment clearly increased the expression of NLRP3 (B) and cleaved caspase-1 (C). $N = 3$ per group. The levels of IL-1 β (D) and IL-18 (E) were analyzed by ELISA. $N = 6$ per group. The representative bands of western blots (F) and quantification the Sirt-1 expression (G) were shown. Treatment with Tet before and after MCAO significantly increased Sirt-1 expression in brain tissues. $N = 3$ per group. Values are mean ± SEM, *$P <$ 0.05 *vs.* Sham; #$P < 0.05$ *vs.* vehicle. Tet, Tetrandrine.

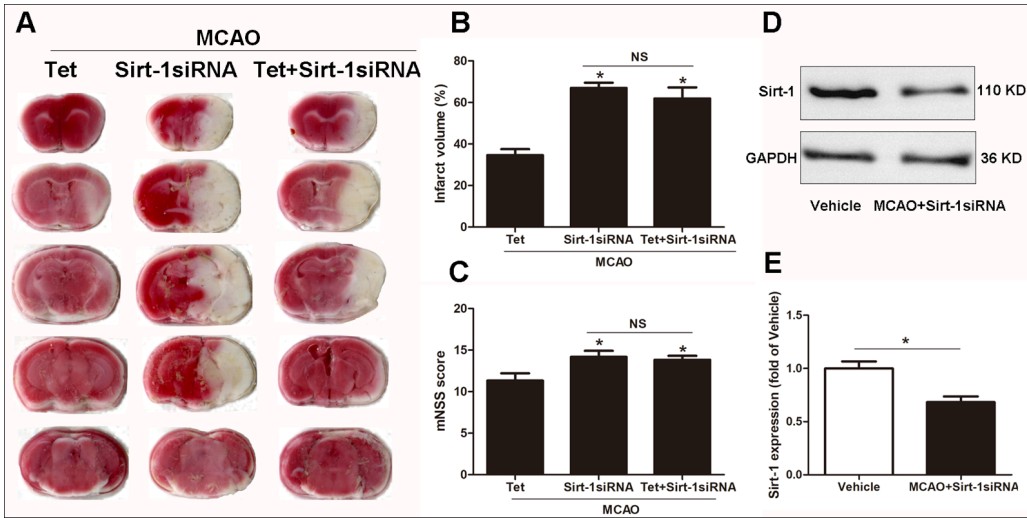

**Figure 4  The effect of Sirt-1 siRNA pretreatment on infarct volumes and neurobehavioral outcomes in cerebral ischemia in mice.** An intracerebroventricular injection of Sirt-1 siRNA was administered to mice to inhibit cerebral Sirt-1 expression, and a scrambled siRNA was injected as a control. (A, B) TTC-stained brain sections showed no significantly different atrophy between Sirt-1 siRNA group and Tet + Sirt-1 siRNA group. $N = 6$ per group, $^*P < 0.05$ *vs.* Tet. (C) Neurological severity scores were detected after cerebral ischemia. $N = 6$ per group. $^*P < 0.05$ *vs.* Tet. (D) Representative western blots showed the protein level of Sirt-1 in mice and was normalized to GAPDH. Sirt-1 siRNA pretreatment significantly inhibited protein level of Sirt-1 in brain tissues. Quantification of the expression level is shown in (E). $N = 3$ per group. $^*P < 0.05$. Values are mean ± SEM. Tet, Tetrandrine. NS, no significance.

findings suggested that Tet downregulated cerebral I/R-induced NLRP3 inflammasome expression through upregulating Sirt-1.

## DISCUSSION

In the present study, we demonstrated that treatment with tetrandrine in mice exerted neuroprotective effects following MCAO. First, it significantly lowered the neurological severity scores and reduced infarct volume and brain edema in the MCAO + Tet group compared with the vehicle group. Second, Tet inhibited NLRP3 inflammasome activation, as evidenced by suppressed expression of NLRP3 in microglial cells and the reduced levels of NLRP3, cleaved caspase-1, IL-1β, and IL-18 in cerebral tissue. Finally, Tet reduced NLRP3-derived inflammation via upregulating Sirt-1, indicating that the protective effects of Tet on cerebral I/R injury in mice were related to Sirt-1 and NLRP3.

Ischemic stroke occurs due to distinctly reduced blood flow to the brain, accompanied by activating the ischemic cascade, which causes to serious neuronal injury. Blood reperfusion is thought to cause more severe secondary tissue damage which induces an inflammatory response to the latter process, leading to additional injury to adjacent brain tissue (*Mizuma & Yenari, 2017*). Unfortunately, few therapy options are available for minimizing tissue damage after a stroke. thus, It is therefore urgent to explore effective agents.

Tet, a natural bisbenzylisoquinoline alkaloid compound, exhibits significant bioavailability and has anti-cancer (*Chen, Chen & Tseng, 2009a*; *Chen, Chen & Tseng,*

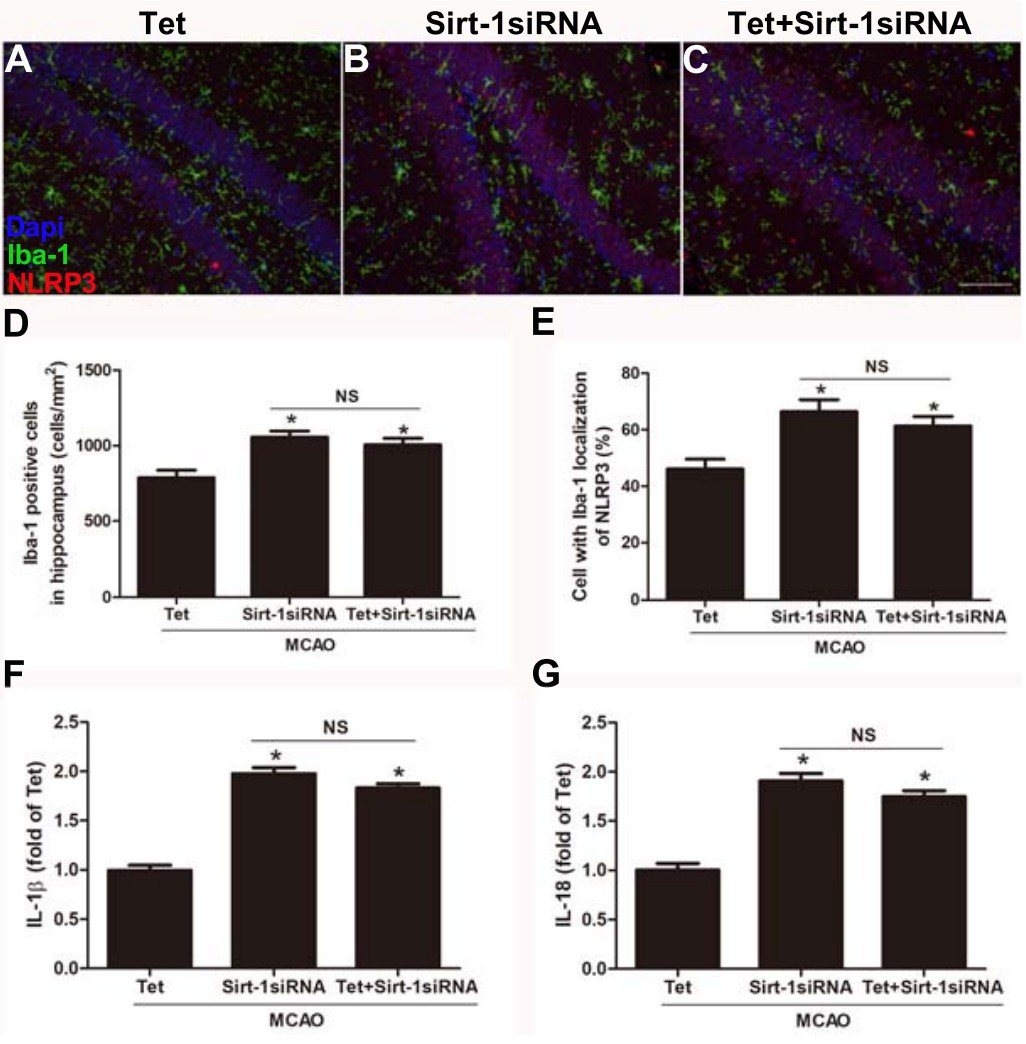

**Figure 5** **Sirt-1 siRNA obviously reverses the effects of Tetrandrine on expression of NLRP3, IL-1 β and IL-18 in mice.** Tet and Sirt-1 siRNA pretreatment blocked the Tet-induced decrease NL RP3 expression in Ib a-1-positive cells in the hippocampus (A–C). Scale bar, 100 μm. Iba-1 positive cells in the hippocampus (D) and percentage of cells with Iba-1 localization of NLRP3 (E) were quantified. Sirt-1 siRNA also blocked the Tet-induced reduction of IL-1 β (F) and IL-18 (G) levels. Values are mean ±SEM, ($n = 6$ per group). *$P < 0.05$ *vs.* Tet. Tet, Tetrandrine. NS, no significance.

*2009b*; *Chen et al., 2009*), anti-inflammatory (*Li et al., 2018*), and cytoprotective effects (*Zhang et al., 2017*). Such pharmacological effects have clinical applications for several conditions, including arrhythmia, silicosis, inflammation, and occlusive cardiovascular disorders (*Chen, Tsai & Tseng, 2011*). Furthermore, Tet has been reported to attenuate heart (*Zhang et al., 2017*), liver (*Liu et al., 2004*), and small bowl (*Chen, Chen & Tseng, 2009a*; *Chen, Chen & Tseng, 2009b*; *Chen et al., 2009*) from I/R injury. *Zhang et al. (2017)* reported that Tet had beneficial effects on I/R-induced injury in cardiac cell models and the mechanisms involved in the JAK3/STAT3/HK II signaling pathway. However, the effects of Tet on ischemic stroke have yet to be determined. Tet has the advantages of being

extremely fat-soluble and hydrophobic and having a low molecular weight, which allow it to cross the blood brain barrier (*Chen, Tsai & Tseng, 2011*). In MCAO mice, Tet (30 mg/kg) contributed to an improvement of cerebral I/R injury partially through regulating GRP78, DJ-1, and HYOU1 protein expression (*Ruan et al., 2013*). In a global cerebral I/R gerbil model, pretreatment with Tet alleviated cortex and hippocampus structural abnormalities (*Sun & Liu, 1995*). In this study, we used C57BL/6 mice subjected to MCAO as an experimental model of focal cerebral I/R injury. We showed that Tet treatment (30 mg/kg/day) for seven consecutive day reduced neurological severity scores, infarct volume, and brain water content. Nevertheless, the mechanisms that underlie ischemic stroke are still not fully understood.

Numerous studies have shown that inflammation is closely associated with the development of cerebral I/R injury (*Meng et al., 2019*; *Mizuma & Yenari, 2017*). NLRP3 inflammasome activation plays a critical pathogenic role in strokes (*Qiu et al., 2016*; *Wang et al., 2015*). The expression of NLRP3 is regarded as a rate-limiting element for inflammasome activation. Generally, the NLRP3 inflammasome is largely located in microglia in the CNS. Previous studies have demonstrated that ischemic stroke can induce microglia activation and promote NLRP3 expression, resulting in neuronal cell death (*Xu et al., 2018*; *Wang et al., 2017*). The activated microglia subsequently release chemokines, cytotoxic mediators, and cytokines, including IL-1β, TNF-α and IL-6, triggering the inflammatory cascade after an ischemic stroke, thus further exacerbating neuroinflammatory damage. It has been shown that downregulation of NLRP3 inflammasome has a neuroprotective effect against ischemic strokes (*Wang et al., 2019a*; *Wang et al., 2019b*). In addition, NLRP3 knockdown by siRNA effectively ameliorated cerebral ischemia damage (*He et al., 2017*). In line with these studies, we found that the percentage of cells with Iba-1 localization of NLRP3 were significantly enhanced in the hippocampus; correspondingly, the levels of NLRP3, cleaved caspase-1, IL-1β, and IL-18 were elevated in the MCAO mice compared with the control group. Above all, these findings suggest that suppressing NLRP3 inflammasome activation might be a beneficial target for ischemic insults.

Tet exhibits anti-inflammatory properties in peripheral reflected by inhibiting T cells, B cells, and the production of cytokines and inflammatory mediators (*Chen, Chen & Tseng, 2009a*; *Chen, Chen & Tseng, 2009b*; *Chen et al., 2009*; *Li et al., 2003*). Furthermore, it has been shown to suppress overexpression of ICAM-1, TNF-α, IL-1 β, and IL-6 in an acute pancreatitis rat model and a transplanted small bowel pig model (*Chen, Chen & Tseng, 2009a*; *Chen, Chen & Tseng, 2009b*; *Chen et al., 2009*; *Wang, Lemos & Iadecola, 2004*). It would thus be worth exploiting the anti-inflammatory activity of Tet after an ischemic stroke. In this study, treatment with Tet drastically suppressed the activation of NLRP3 inflammasome in the MCAO group.

To understand how Tet inhibits the NLRP3 inflammasome, we focused on protein deacetylase Sirt-1 because of its high expression in the CNS. Importantly, Sirt-1 plays a critical role in neuroprotective mechanisms against cerebral ischemic injury and exhibits anti-inflammatory activity through mediating NLRP3 inflammasome activation (*Yang et al., 2015*; *Ma et al., 2015*). In our study, Tet treatment remarkably increased Sirt-1 expression in MCAO mice. Intriguingly, Sirt-1 knockdown using siRNA significantly

reduced the neuroprotective effects and reversed the suppression of NLRP3 inflammasome activation mediated by Tet. Our results indicate that Tet exerts neuroprotective effects against ischemic stroke injury partly through inhibiting the activation of the NLRP3 inflammasome via upregulating Sirt-1.

## CONCLUSIONS

Our results showed that treatment with Tet can protect against cerebral I/R-induced brain tissue injury in mice, which is possibly associated with the suppression of Sirt-1-mediated NLRP3 inflammasome activation. Our study demonstrates the potential of Tet for ameliorating cerebral I/R injury, suggesting its clinical advantages for cardiovascular disease therapy. Nevertheless, further investigations are needed to understand more precisely the mechanism underlying Tet with neuroinflammatory and its possible applications in ischemic stroke therapy.

### Funding
The authors received no funding for this work.

### Competing Interests
The authors declare there are no competing interests.

### Author Contributions
- Jun Wang performed the experiments, analyzed the data, prepared figures and/or tables, and approved the final draft.
- Ming Guo conceived and designed the experiments, analyzed the data, prepared figures and/or tables, authored or reviewed drafts of the paper, and approved the final draft.
- Ruojia Ma and Maolin Wu performed the experiments, prepared figures and/or tables, and approved the final draft.
- Yamei Zhang conceived and designed the experiments, analyzed the data, authored or reviewed drafts of the paper, and approved the final draft.

### Animal Ethics
The following information was supplied relating to ethical approvals (i.e., approving body and any reference numbers):

The experimental procedures were approved by the Ethics Committee of Laboratory Animal Care and Welfare, Zhejiang Academy Medical Sciences, with the proved number 2018-143.

### Data Availability
The raw measurements are available in the Supplemental Files.

## Supplemental Information

Supplemental information for this article can be found online at http://dx.doi.org/10.7717/peerj.9042#supplemental-information.

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
