# Peer review of "Tetrandrine alleviates cerebral ischemia/reperfusion injury by suppressing NLRP3 inflammasome activation via Sirt-1"

_PeerJ, doi:10.7717/peerj.9042_

## Round 0.1 · original submission · Major Revisions

Your manuscript has been revised by three experts and they recommended major revision. Note that the inclusion of new data will be imperative ( How silenced efficient of Sirt-1 siRNA in vivo., please provide experimental data). This point was emphasized by two reviewers. Note that there are other problems with the data ( However, the design of the animal experiments is confused as the raw data is not identical to the presented figure). There are also problems with the figures and possibly with their data. In addition to these points, there are several other important aspects raised by the 3 reviewers that you have to correct or explain in a better way.

·

Basic reporting

1. Discussion should be added to existing relevant reports.
Eg.
[1] Zhang Juntie, Guo Ruixia, Li Xia, et al. Tetrandrine Cardioprotection in Ischemia-Reperfusion (I/R) Injury via JAK3/STAT3/Hexokinase II [J]. Eur J Pharmacol, 2017, 813(1): 153-160. DOI: 10.1016/j.ejphar.2017.08.019

Experimental design

1. “Materials & Methods – Experimental protocols” Line 144 to 146. Please provide references on dosage and method of Tetrandrine treatment.

Validity of the findings

1. “Materials & Methods – Experimental protocols” Line 141-143. “A total of 45 mice were randomly divided into three groups: sham operation (Sham, n = 15), MCAO treatment with vehicle (Vehicle, n = 15) or tetrandrine (MCAO+Tet, n = 15).” In Figure 1B, it was shown five brain sections in each groups, but in supplemental file “File_3._Raw_data-_Fig1.”, it was shown six brain infarct volume data in each groups! Therefore, how many mice were in each group, and why the number did not match!
2. “Materials & Methods – Intracerebroventricular injection of Sirt-1 siRNA” Line 194-203. What is the siRNA transfection method used? The method should be described more detailed. How silenced efficient of Sirt-1 siRNA in vivo., please provide experimental data.

Additional comments

Ischemic stroke is a leading cause of disability and death worldwide, which creates a heavy burden to patients and to society in the long run. Tetrandrine (TET), a bisbenzylisoquinoline alkaloid has been used for the treatment of cardiovascular diseases and hypertension. Recent studies provide the TET exerts cardioprotection in ischemia-reperfusion (I/R) injury, but the mechanisms was not well clear. In this manuscript, Dr. Wang and colleagues looked at the possible actions of the TET on cerebral ischemia and the related mechanisms involved in NLRP3. The study’s conclusion showed that TET has benefits for cerebral I/R injury partially related to suppression of NLRP3 inflammasome activation via regulating Sirt-1. Although the current study is interesting, there are some major concerns that need to be addressed before consideration of publication.

Major concerns:
1. In the 2017 published article [1], Dr. Zhang and colleagues also analyzed the mechanisms of tetrandrine exerts cardioproteciton in ischemia-reperfusion (I/R) injury, so it is necessary to discuss the similarities and differences between the two studies.
[1] Zhang Juntie, Guo Ruixia, Li Xia, et al. Tetrandrine Cardioprotection in Ischemia-Reperfusion (I/R) Injury via JAK3/STAT3/Hexokinase II [J]. Eur J Pharmacol, 2017, 813(1): 153-160. DOI: 10.1016/j.ejphar.2017.08.019
2. “Materials & Methods – Experimental protocols” Line 141-143. “A total of 45 mice were randomly divided into three groups: sham operation (Sham, n = 15), MCAO treatment with vehicle (Vehicle, n = 15) or tetrandrine (MCAO+Tet, n = 15).” In Figure 1B, it was shown five brain sections in each groups, but in supplemental file “File_3._Raw_data-_Fig1.”, it was shown six brain infarct volume data in each groups! Therefore, how many mice were in each group, and why the number did not match!
3. “Materials & Methods – Intracerebroventricular injection of Sirt-1 siRNA” Line 194-203. What is the siRNA transfection method used? The method should be described more detailed. How silenced efficient of Sirt-1 siRNA in vivo., please provide experimental data.
4. “Results – Tet inhibited the NLRP3 inflammasome-derived inflammation in cerebral I/R injury in mice” Line 245, Figure 3A to C. The western bolting should show the all samples in each group.
5. “Results – Tet inhibited the NLRP3 inflammasome-derived inflammation in cerebral I/R injury in mice” Line 245-246, Figure 3D and E, ELISA is the quantitative analysis, so the “IL-1beta and IL-18” protein levels should be shown by absolute value.
6. “Results – Tet upregulated Sirt-1 expression and suppressed NLRP3 inflammasome activation through Sirt-1” Line 256, Figure 3F and G. The western bolting should be shown the all samples in each group.
7. “Results – Tet upregulated Sirt-1 expression and suppressed NLRP3 inflammasome activation through Sirt-1” Line 257-259. “…cerebral Sirt-1 was partially knocked down by siRNA in mice.” Please provide experimental data.
8. “Results – Tet upregulated Sirt-1 expression and suppressed NLRP3 inflammasome activation through Sirt-1” Line 263-265, Figure 5. “…Sirt-1 siRNA markedly enhanced expression of the NLRP3…” Immunofluorescence data are not sufficient to support this conclusion, so western bolting should be perform same as Figure 3A to C.
9. “Results – Tet inhibited the NLRP3 inflammasome-derived inflammation in cerebral I/R injury in mice” Line 266, Figure 5B and C, the protein levels of IL-1beta and IL-18 should be shown by absolute value.

Minor comments:
1. “Materials & Methods – Experimental protocols” Line 144 to 146. Please provide references on dosage and method of Tetrandrine treatment.
2. “Results – Tet inhibited the NLRP3 inflammasome-derived inflammation in cerebral I/R injury in mice” Line 245, Figure 3A and C. “Caspase-1” should be marked by “Cleaved caspase-1”.

Reviewer 2 ·

Basic reporting

Language
1.The English writing is also necessary to be revised and polished again before it is accepted. It needs a minor revision to improve the scientific quality for publication.
Literature references
1.In the introduction section (line 95-97), the authors wrote " Tet has received extensive
attention for its anti-tumor, anti-inflammatory, and analgesic actions". This sentence
needs referencing.
2. Line 147-148. The authors wrote " Brains were harvested for the subsequent
experiments (Figure 1A).". This part should belong to the results.
3. In the line 304, the authors wrote " such as anti-cancer, anti-inflammatory, and
cytoprotective effects.". This sentence needs concrete referencing.
4. In the line 307, the authors wrote " Tet has been reported to attenuate heart, liver and
small bowl from I/R injury.". This sentence needs concrete referencing.
Figure
1. Figure 3A is very small and of poor quality and cannot be evaluated.
2.Please explain why the expression of caspase-1 is high in the group of sham
in Fig 3A.
3.Fig 3F is almost invisible.
4.Please supplement the relative molecular weight of all protein antibodies in Fig 3A and
3F.
5. Fig4A the fifth piece in the group of Sirt-1 siRNA is incomplete and the third piece in the
group of Tet+Sirt-1 siRNA is problematic.
6.In Fig 5A, NLRP3-positive cells in Iba-1-positive cells in hippocampus should be
quantified and compared between the groups.
7. In Fig 5A, all images should be taken from the same field of view, and the author should
offer the scale bar.

Experimental design

The author firstly demonstrated that Tetrandrine alleviates cerebral ischemia/reperfusion injury by suppressing NLRP3 inflammasome activation via Sirt-1. The idea is interesting and novel.The logic thinking of results in this article , however, needs should be considered.
1.Please illustrate the reasons selected a concentration of 30 mg/kg tetrandrine
intraperitoneally once a day for 7 days, and supplement more data to accord with the
choice.
2.The logic thinking of results in this article needs should be considered.

Validity of the findings

no comment

Additional comments

The author should know that the research articles must be carefully written.

·

Basic reporting

no comment

Experimental design

no comment

Validity of the findings

no comment

Additional comments

In this manuscript, the authors demonstrated that Tetrandrine(Tet) reduces cerebral I/R injury through inhibition NLRP3 inflammasome activation by upregulating Sirt-1. They established a cerebral I/R injury mice model through MCAO surgery to confirm Tet had the neuroprotective affects. Furthermore, they indicated that Tet upregulated Sirt-1 and sequentially inhibited NLRP3 inflammasome activation. This study aims to uncover the biological function and potential molecular mechanism of Tet in cerebral I/R injury. However, there are several concerns in the whole project.
1. The description “treatment of Tet could reduce the neuro- injury in MCAO model of mice” has already concluded from previous study in introduction section. So, the novelty of the whole study is not good enough at least in the first part of study.
2. The effect of Sirt-1 siRNA should be detected and shown before in vivo experiments.
3. In Materials & Methods, 45 mice were randomly divided into 3 groups to detect the function of Tet and Sirt-1 siRNA in cerebral I/R injury mice. However, the design of the animal experiments is confused as the raw data is not identical to the presented figure.
4. The authors described that Sirt-1 siRNA was injected into the right lateral ventricle. What about the control group (Tet group) in Figure 5? Scientifically, they should be carried out the same operating procedure with the solvent which was used to dilute Sirt-1 siRNA. So, the manipulations of study should be conducted rigorously.
5. In Abstract, Results section, I think the authors mean “Tet attenuated I/R-induced enhanced expression of NLRP3 inflammasome, caspase-1, IL-1β, IL-18 via upregulating Sirt-1”, not “and Sirt-1”.
6. The levels of IL-1β and IL-18 by ELISA assay were presented in relative fold change to sham group, please provided the absolute quantitative level of them in each sample.
7. The quantitative data and method of Immunofluorescence assay in Figure 2 and Figure 5 should be provided.

---

## Round 0.2 · accepted · Accept

Thank you for revising your manuscript. The reviewers have now recommended its publication.

·

Basic reporting

No comment.

Experimental design

No comment.

Validity of the findings

No comment.

Additional comments

Ischemic stroke is a leading cause of disability and death worldwide, which creates a heavy burden to patients and to society in the long run. Tetrandrine (TET), a bisbenzylisoquinoline alkaloid has been used for the treatment of cardiovascular diseases and hypertension. Recent studies provide the TET exerts cardioprotection in ischemia-reperfusion (I/R) injury, but the mechanisms was not well clear. In this manuscript, Dr. Wang and colleagues looked at the possible actions of the TET on cerebral ischemia and the related mechanisms involved in NLRP3. The study’s conclusion showed that TET has benefits for cerebral I/R injury partially related to suppression of NLRP3 inflammasome activation via regulating Sirt-1. The paper is significantly improved and all concerned raised by the reviewer have been addressed. I think it is suitable for publication at this point for this version of revised manuscript.

·

Basic reporting

No comment.

Experimental design

No comment.

Validity of the findings

No comment.

Additional comments

In this manuscript, Dr. Wang and colleagues demonstrated that Tetrandrine (Tet) reduces cerebral I/R injury through inhibition NLRP3 inflammasome activation by upregulating Sirt-1. They established a cerebral I/R injury mice model through MCAO surgery to confirm Tet had the neuroprotective affects. Furthermore, they indicated that Tet upregulated Sirt-1 and sequentially inhibited NLRP3 inflammasome activation. This study aims to uncover the biological function and potential molecular mechanism of Tet in cerebral I/R injury. The authors have revised the paper carefully according to the concerns raised by reviewers in the current version, and I think it is suitable for publication.